# EfficientVLA: Training-Free Acceleration and Compression for Vision-Language-Action Models

**Yantai Yang**[1,2] **Yuhao Wang**[1,3] **Zichen Wen**[1] **Luo Zhongwei**[1] **Chang Zou**[1,4] **Zhipeng Zhang**[1,5]
**Chuan Wen**[1]* **Linfeng Zhang**[1]*

[1]School of Artificial Intelligence, Shanghai Jiao Tong University
[2]Harbin Institute of Technology
[3]Xi'an Jiaotong University
[4]University of Electronic Science and Technology of China
[5]Anyverse Dynamics

 **Code:** https://github.com/YantaiYang-05/EfficientVLA

## Abstract

Vision-Language-Action (VLA) models, particularly diffusion-based architectures, demonstrate transformative potential for embodied intelligence but are severely hampered by high computational and memory demands stemming from extensive inherent and inference-time redundancies. While existing acceleration efforts often target isolated inefficiencies, such piecemeal solutions typically fail to holistically address the varied computational and memory bottlenecks across the entire VLA pipeline, thereby limiting practical deployability. We introduce **EfficientVLA**, a structured and training-free inference acceleration framework that systematically eliminates these barriers by cohesively exploiting multifaceted redundancies. EfficientVLA synergistically integrates three targeted strategies: (1) pruning of functionally inconsequential layers from the language module, guided by an analysis of inter-layer redundancies; (2) optimizing the visual processing pathway through a task-aware strategy that selects a compact, diverse set of visual tokens, balancing task-criticality with informational coverage; and (3) alleviating temporal computational redundancy within the iterative diffusion-based action head by strategically caching and reusing key intermediate features. We apply our method to a standard VLA model CogACT, yielding a $1.93\times$ inference speedup and reduces FLOPs to $28.9\%$, with only a $0.6\%$ success rate drop in the SIMPLER benchmark.

## 1 Introduction

Building upon advances in multimodal understanding from models integrating vision and language [1, 2, 3, 4, 5], Vision-Language-Action (VLA) models enable transformative embodied intelligence. These systems, such as OpenVLA [6], CogACT [7], $\pi_0$ [8] and RT-2 [? ], directly translate multimodal inputs into executable actions, successfully tackling complex robotic manipulation and reasoning tasks using large-scale datasets [9, 10]. Many cutting-edge VLAs couple a Vision-Language Model (VLM) for scene and instruction parsing with a diffusion model to handle multi-modal action distribution [7, 11, 12, 13]. However, the significant computational and memory overheads of these Diffusion-based VLA architectures during the inference time pose critical barriers to their practical deployment, particularly for real-time interaction on resource-constrained robotic platforms.

Diffusion-based VLA architectures typically comprise a vision encoder to extract features, a large language model (LLM) [14, 15, 16, 17] core for multimodal reasoning, and a diffusion-based action

---

*Corresponding authors.

Table 1: Module-wise inference characteristics of a baseline VLA model (CogACT, Left) compared to our proposed EfficientVLA (Right). EfficientVLA demonstrates significant improvements in overall inference speed and computational efficiency (FLOPs).

| | Vision Module | Language Module | Action Module | | Vision Module | **Language Module** | **Action Module** |
|---|---|---|---|---|---|---|---|
| #Param (M) | 802.3 | 6738.9 | 89.0 | #Param (M) | 802.3 | **3971.1** ($\downarrow$41%) | 89.0 |
| Vision Token | 256 | 256 | - | Vision Token | 256 | **56** ($\downarrow$78%) | - |
| Denoising Steps | - | - | 10 | Denoising Steps | - | - | **2** ($\downarrow$80%) |
| Inference Time (ms) | 24.9 | 134.5 | 51.5 | Inference Time (ms) | 24.9 | **58.9** ($\downarrow$56%) | **26.2** ($\downarrow$49%) |
| FLOPs (G) | 405.50 | 3726.55 | 57.96 | FLOPs (G) | 405.50 | **792.58** ($\downarrow$78%) | **11.72** ($\downarrow$80%) |

(a) Token-wise inference bottlenecks  (b) Layer-wise output cosine similarity  (c) Timestep-wise output cosine similarity

Figure 1: VLA inference bottleneck and redundancy analysis: **(a)** Visual token pruning impact on FLOPs and inference time, revealing computation-bound and memory-bound regimes. **(b)** High inter-layer cosine similarity of LLM hidden states, indicating depth-wise redundancy. **(c)** Temporal cosine similarity of MLP/attention features in diffusion steps, showing computational redundancy.

decoder to predicts the final actions through multiple denoising steps. While this modular design underpins their powerful capabilities, it inherently results in substantial computational and memory overhead. Our findings (Table 1) indicate that the language module and the iterative diffusion head are primary contributors to overall latency and computational load. Furthermore, as illustrated in Figure 1 (a), while visual token pruning initially reduces inference time in computation-bound scenarios, its efficacy quickly diminishes as the system becomes memory-bound by the LLM.

Prior VLA acceleration efforts have largely focused on isolated tweaks, delivering minimal overall gains. These fragmented approaches often fail because they ignore the integrated nature of VLA, where optimizing one module in isolation merely shifts bottlenecks. Gains are limited by unaddressed inefficiencies elsewhere, such as the memory demands of LLM or the computational intensity of action head. For example, methods like TinyVLA [13] and DeeR-VLA [18] focus on specialized model architectures rather than broadly applicable inference acceleration frameworks for pre-trained VLAs. Other approaches, such as Mole-VLA [19], tackle LLM layer redundancy but require costly retraining and overlook other pipeline stages. Similarly, VLA-Cache [20] caches static visual tokens but provides limited speedup, constrained by the significant memory footprint of LLM and computational demands of the action head. Consequently, these existing approaches fall short of providing a truly holistic solution to navigate the complex landscape of VLA inefficiencies.

To develop a more effective acceleration strategy, we systematically analyze the inference characteristics and multifaceted redundancies within each VLA module. In many Diffusion-based VLAs, the diffusion action head operates as a separate module, guided by features extracted from the VLM. This separation may underutilize the full reasoning capacity of VLM for action generation, questioning the necessity of its entire scale. As illustrated in Figure 1 (b), the language module demonstrates shows considerable depth-wise representational redundancy with high inter-layer hidden state similarity. The visual processing pathway exacerbates this issue by processing superfluous tokens, characterized by low task-relevance or high informational overlap due to visual similarity, which strains computational resources and intensifies the memory-bound condition of LLM. As shown in Figure 1 (c), the iterative diffusion action head displays significant temporal redundancy. The high similarity of its intermediate features across adjacent denoising steps implies extensive and near-static recomputations.

Motivated by this, we introduce EfficientVLA, a structured, training-free acceleration framework for Diffusion-based VLAs that systematically targets these issues. Using a similarity-derived importance metric to target the primary memory bottleneck of the language module and its observed depth-wise

redundancy (Figure 1 (b)), EfficientVLA employs a similarity-derived importance metric to prune functionally inconsequential layers, thus reducing the depth of the model and the demands for memory without retraining. To manage the initial computational load from visual inputs before the memory of LLM limit is reached (Figure 1 (a)), our visual token pruning strategy tackles both task-relevant and inherent image redundancies by first selecting critical task-aligned tokens, then augmenting this set to ensure representational diversity while maintaining high task relevance. Lastly, EfficientVLA addresses temporal redundancy in the compute-intensive action generator (highlighted by high feature similarity across timesteps, Figure 1 (c)) by caching and reusing intermediate attention and MLP outputs, thus curtailing redundant computations. This synergistic, structured approach provides a more holistic alleviation of GPU compute and memory bottlenecks than isolated optimizations.

The main contributions of this work are summarized as follows:

1. We present a systematic analysis identifying critical computational and memory-bound bottlenecks, alongside multifaceted redundancies within contemporary Diffusion-based Vision-Language-Action (VLA) architectures, thereby motivating the need for structured acceleration.

2. We propose *EfficientVLA*, a novel training-free, structured inference acceleration framework that synergistically prunes redundant layers from the language module based on their informational impact and strategically selects a compact, task-focused subset of visual tokens by considering both VLA task relevance and inherent image feature diversity.

3. Our framework further enhances efficiency by exploiting temporal redundancies in the diffusion-based action head, introducing a caching mechanism for intermediate attention and MLP computations during the iterative denoising process.

4. We demonstrate the efficacy of EfficientVLA through extensive experiments on the CogACT in the SIMPLER environment [21], achieving a $1.93\times$ inference speedup and reducing FLOPs to $28.9\%$, all while incurring a minimal accuracy degradation of only $0.6\%$. This will facilitate the application of large-scale VLAs on the resource-constrained robotics platforms in the real world.

## 2 Related Work

**Vision-Language-Action (VLA) Models**. Vision-Language-Action models [6, 18, 22, 23, 24, 25, 26, 27] extend Vision-Language Models (VLMs) [3, 28, 29, 30, 31] by incorporating action generation, bridging the gap between perception and action. These models enable machines to understand visual and textual inputs and generate corresponding actions for tasks [22, 32] such as robotic manipulation and object retrieval. VLA models typically use pretrained VLMs [29] to encode visual and linguistic data into a shared representation, from which actions are generated either as discrete tokens or continuous values. A prominent recent trend within VLA is the adoption of diffusion models for generating coherent continuous action sequences. This paradigm is exemplified by models such as CogACT [7], DexVLA [12], DiVLA [11], $\pi_0$ [8], and TinyVLA [13]. Many of these diffusion-based VLAs employ a componentized design: the foundational VLM processes visual and linguistic inputs to produce a condensed feature representation, which then conditions a distinct diffusion-based action module responsible for the iterative generation of precise action trajectories. This often involves the VLM output steering the denoising process within the specialized action decoder.

**Efficient Vision-Language-Action Models**. The computational complexity of VLMs [33, 34, 35, 36] poses significant challenges for their real-time deployment, particularly in applications such as robotic control that require rapid decision-making. To address this issue, recent efforts to accelerate VLA models have been primarily categorized into training-aware and training-free methods. Training-aware approaches, such as RoboMamba [37], EfficientVLM [38], and DeeR-VLA [18], focus on optimizing model architectures or applying compression techniques followed by retraining, achieving significant speedups while maintaining performance. For instance, DeeR-VLA reduces computational costs by leveraging dynamic reparameterization and efficient pruning strategies, which enable more flexible and scalable model deployment. Similarly, For example, Mole-VLA [19] reduces computational costs by dynamically activating only a subset of model layers based on task-specific needs. In contrast, training-free methods, such as VLA-Cache [20], enhance efficiency by reusing previously computed results for unchanged tokens between consecutive frames, which is particularly beneficial in scenarios with minimal variation in visual input.

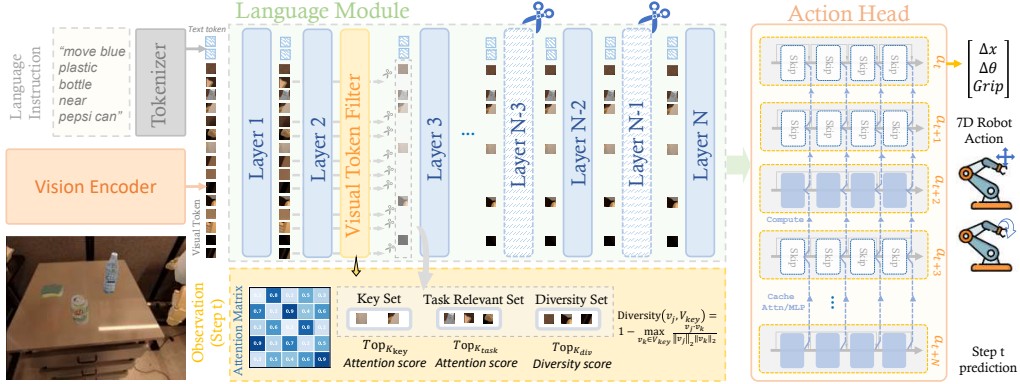

Figure 2: Overview of the **EfficientVLA** framework, our training-free, structured approach to accelerate Diffusion-based VLAs. It employs: (1) pruning of redundant language module layers; (2) VLA task-aware visual token selection balancing task relevance and informational diversity; and (3) temporal caching of intermediate featuresin the diffusion action head.

# 3 Method

## 3.1 Preliminaries: Vision-Language-Action Models

Vision-Language-Action (VLA) models represent a class of multimodal systems designed to bridge perception, language understanding, and robotic action. These models typically process image observations and natural language instructions through a sequence of specialized modules to generate executable action sequences. The initial stage of our basic VLA model employs a Vision Module, comprising powerful pre-trained encoders DINOv2 [39] and SigLIP [40], to transform the raw visual input $O_{img}$ into a set of rich feature embeddings $F_V$. These visual features $F_V$, along with tokenized language instructions, are then ingested by a language model backbone. This LLM performs multimodal fusion and contextual reasoning to derive a task-oriented representation or conditioning signal, $F_{VL}$, which encapsulates the understanding of the scene and the instructed goal. Finally, a Diffusion-based Action Head takes the cognition feature extracted from the output feature $F_{VL}$ as input and predicts the final action space of a gripper with 7 degrees of freedom (DoF).

## 3.2 Vision-Language Model Pruning

### 3.2.1 Layer Redundancy Analysis

The language module within VLA models, typically a multi-layer Transformer decoder, is critical for multimodal reasoning but often introduces substantial computational overhead. Each layer $\ell$ in such a transformer updates its input hidden state $\boldsymbol{x}^{(\ell)} \in \mathbb{R}^{d \times S}$ via a residual transformation: $\boldsymbol{x}^{(\ell+1)} = \boldsymbol{x}^{(\ell)} + f(\boldsymbol{x}^{(\ell)}, \theta^{(\ell)})$, where $f(\cdot)$ is the layer-specific function with parameters $\theta^{(\ell)}$, $d$ is the hidden dimension, and $S$ is the sequence length. Our empirical analysis, illustrated in Figure 1 (b), reveals significant depth-wise representational redundancy within this language module component. Specifically, we observe high cosine similarity between the input $\boldsymbol{x}^{(\ell)}$ and output $\boldsymbol{x}^{(\ell+1)}$ states for numerous, particularly deeper layers. This indicates that the effective transformation $f(\boldsymbol{x}^{(\ell)}, \theta^{(\ell)})$ imparted by these layers is minimal, rendering them functionally less critical and prime candidates for pruning to enhance inference efficiency with negligible impact on task performance.

### 3.2.2 Importance-Driven Non-Contiguous Layer Pruning

To address the identified depth-wise redundancy within the language module of VLA models, we first rigorously quantify the functional importance of each layer. Our approach aims to identify layers that contribute minimally to the transformation of hidden state representations, rendering them candidates for pruning. We define the importance score $I^{(\ell)}$ for a given layer $\ell$ based on the principle that a layer effecting substantial change to its input is more critical than one whose output closely mirrors its input. Specifically, $I^{(\ell)}$ is quantified as one minus the average cosine similarity between its input and output hidden states across a representative dataset $\mathcal{D}$ of VLA training samples and all $L$ token

positions within each sample:

$$I^{(\ell)} = 1 - \frac{1}{|\mathcal{D}|} \sum_{i=1}^{|\mathcal{D}|} \left( \frac{1}{L} \sum_{j=1}^{L} \frac{\boldsymbol{x}_{i,j}^{(\ell)} \cdot \boldsymbol{x}_{i,j}^{(\ell+1)}}{\|\boldsymbol{x}_{i,j}^{(\ell)}\|_2 \|\boldsymbol{x}_{i,j}^{(\ell+1)}\|_2} \right) \tag{1}$$

where $\boldsymbol{x}_{i,j}^{(\ell)}, \boldsymbol{x}_{i,j}^{(\ell+1)} \in \mathbb{R}^d$ denote the input and output hidden state vectors, respectively, at position $j$ of sample $i$ for layer $\ell$. A high cosine similarity signifies a minimal transformative effect by the layer function $f(\boldsymbol{x}^{(\ell)}, \theta^{(\ell)})$, resulting in a low importance score $I^{(\ell)}$ and indicating functional redundancy.

Based on these importance scores, we employ a non-contiguous pruning strategy. For an LLM comprising $N$ layers, the importance score $I^{(\ell)}$ is computed for every layer $\ell \in \{1, \ldots, N\}$. These scores are then sorted in ascending order, yielding an ordered list of layer indices $\mathcal{L}_{ranked} = [\ell_{(1)}, \ell_{(2)}, \ldots, \ell_{(N)}]$ such that $I^{(\ell_{(1)})} \le I^{(\ell_{(2)})} \le \cdots \le I^{(\ell_{(N)})}$. Subsequently, the first $n$ layers from this list, $\{\ell_{(1)}, \ell_{(2)}, \ldots, \ell_{(n)}\}$, are selected for removal from the model.

### 3.3 Task-Relevance and Diversity-Driven Visual Token Pruning

Visual token streams processed by VLA models, despite their rich informational content, frequently exhibit significant redundancy, imposing substantial computational and memory overhead. This redundancy typically manifests in two primary forms: (i) tokens possessing low relevance to the specific VLA task objectives and (ii) tokens that are informationally duplicative due to inherent visual similarities within the input. To counteract these distinct forms of superfluity, we introduce a novel, training-free, VLA task-aware visual token pruning methodology. Our approach strategically distills a compact yet maximally informative subset of visual tokens, $V_{pruned} \subset V$ of a predetermined size $K_{final}$ (from an initial set of $N_{total}$ token embeddings $V = \{v_1, v_2, \ldots, v_{N_{total}}\}$ derived from the input image). This is achieved by first anchoring the selection with task-critical tokens identified via attention analysis, and subsequently augmenting this core set by judiciously balancing continued task relevance with the explicit promotion of feature diversity through similarity measures. The retained visual tokens for inference can be found in the supplementary material.

#### 3.3.1 Quantifying Task Relevance

To guide visual token pruning, we quantify the task relevance for each initial visual token $v_i$ (from a set of $N_{total}$) by leveraging cross-attention scores from selected VLM layers. These scores capture the attention from $v_i$ towards $L_{ctx}$ task-defining contextual embeddings (e.g., language instructions). Let $A_{i,j}^{(h)}$ denote the attention from visual token $v_i$ to the $j^{th}$ contextual token in the $h^{th}$ attention head (of $H$ total heads). The raw task relevance score $r_i$ for $v_i$ is computed by first averaging attention contributions across all $H$ heads for each visual-contextual pair $(i, j)$, and then summing these averaged attentions over all $L_{ctx}$ contextual elements:

$$r_i = \sum_{j=1}^{L_{ctx}} \left( \frac{1}{H} \sum_{h=1}^{H} A_{i,j}^{(h)} \right) \tag{2}$$

These raw scores $r_i$, signifying each token's overall engagement with the task context, are subsequently normalized (e.g., via min-max scaling) to standardized scores $s_i \in [0, 1]$ for robust comparison and subsequent token selection.

#### 3.3.2 Selection of Key Task-Relevant Tokens

Armed with the normalized task relevance scores $\{s_i\}$, the first phase of pruning identifies an initial set of $K_{key}$ visual tokens (e.g., $K_{key}$ empirically set between 4 and 8) that demonstrate the highest relevance to the VLA task. These tokens constitute the core and indispensable visual token set, $V_{key}$:

$$V_{key} = \{v_i \in V \mid s_i \text{ is among the top } K_{key} \text{ scores in } \{s_k\}_{k=1}^{N_{total}}\} \tag{3}$$

The tokens in $V_{key}$ are unconditionally retained in $V_{pruned}$, forming a foundational scaffold of visual cues deemed essential for task comprehension and successful execution. The set of remaining candidate tokens for further consideration is denoted as $V_{rem} = V \setminus V_{key}$.

### 3.3.3 Augmentative Selection Balancing Relevance and Diversity

To supplement the core set $V_{key}$ and achieve the target final token count $K_{final}$, an additional $K_{aug} = K_{final} - K_{key}$ tokens are meticulously selected from $V_{rem}$. This crucial augmentation phase is guided by a ratio $\alpha \in [0, 1]$, which orchestrates a hybrid selection strategy that concurrently promotes continued emphasis on task relevance and the introduction of informational diversity.

**Task-Driven Augmentation.** A fraction of the augmentation quota, specifically $K_{task} = \lfloor \alpha \cdot K_{aug} \rfloor$ tokens, is selected from $V_{rem}$ by further prioritizing tokens based on their high task relevance scores $s_i$. $V_{task}$ reinforces the task-centric nature of the pruned representation by incorporating additional tokens that, while not part of the initial $K_{key}$ elite, still exhibit strong relevance signals. These tokens are added to the selection, and the pool of remaining candidates is updated: $V_{rem} \leftarrow V_{rem} \setminus V_{task}$.

**Diversity-Driven Augmentation.** The remaining $K_{div} = K_{aug} - K_{task}$ tokens are selected from the updated $V_{rem}$ with the explicit objective of maximizing feature diversity relative to the key selected tokens. This step is vital for capturing a broader spectrum of visual information and mitigating inherent redundancies not addressed by task relevance alone. For each candidate token $v_j \in V_{rem}$, its dissimilarity to the set $V_{key}$ is computed. A common measure is the cosine distance, ensuring that selected tokens are distinct in the embedding space:

$$\text{Diversity}(v_j, V_{key}) = 1 - \max_{v_k \in V_{key}} \frac{v_j \cdot v_k}{\|v_j\|_2 \|v_k\|_2} \tag{4}$$

The $K_{div}$ tokens from $V_{rem}$ exhibiting the highest dissimilarity scores (i.e., those maximally different from already selected tokens) are chosen to form the set $V_{div}$. This targeted inclusion of diverse tokens ensures the final selection is not overly specialized and retains a richer contextual understanding.

**Final Pruned Visual Token Set.** The comprehensive set of visual tokens retained after pruning is the union of these strategically selected components:

$$V_{pruned} = V_{key} \cup V_{task} \cup V_{div} \tag{5}$$

This final set $V_{pruned}$, of cardinality $K_{final}$, is subsequently utilized for all downstream processing within the VLA model. This systematic reduction in visual sequence length significantly alleviates computational demands while preserving critical task-specific and diverse visual information.

## 3.4 Caching Intermediate Features in Action Prediction

Generating high-fidelity action sequences with Diffusion-based VLA models involves an iterative denoising process that demands significant computation due to repeated self-attention and MLP computations over $T$ timesteps. We observe strong temporal coherence in the intermediate features produced during action generation (Figure 1 (c)), indicating substantial redundancy across timesteps. To address this inefficiency and accelerate the action generation phase, we propose a static caching mechanism. This strategy periodically recomputes and caches critical intermediate attention and MLP output at a fixed interval $N$, reusing these cached values for the intervening time steps in the generation of action sequences. This selective computation aims to significantly reduce the computational cost associated with generating the action sequence while preserving its quality.

### 3.4.1 Feature Generation and Temporal Coherence in DiT Blocks

Let $t$ denote the current denoising timestep, typically iterating from an initial $T_{start}$ down to 1. Within each DiT block at timestep $t$, the input features $\mathbf{z}_t$ (which may incorporate cognitive features $\mathbf{f}_t$ from upstream VLM modules and the current noise estimate) are processed sequentially by a self-attention module and an MLP module to produce intermediate hidden states:

$$\mathbf{h}_t^{\text{attn}} = \text{Self-Attn}(\mathbf{z}_t) \tag{6}$$

$$\mathbf{h}_t^{\text{mlp}} = \text{MLP}(\mathbf{h}_t^{\text{attn}} + \mathbf{z}_t) \tag{7}$$

These features, $\mathbf{h}_t^{\text{attn}}$ and $\mathbf{h}_t^{\text{mlp}}$, are fundamental to the denoising capacity of the diffusion model. Our observation of their high temporal coherence—meaning $\mathbf{h}_t^{\text{module}} \approx \mathbf{h}_{t-1}^{\text{module}}$ for many $t$ and module types—motivates their periodic caching and reuse.

Table 2: Performance of EfficientVLA on the CogACT versus the other baselines in the SIMPLER environment. Settings vary by retained LLM layers (L) and visual tokens (T). *Random Dropping* denotes a method involving the random retention of 112 visual tokens.

| SIMPLER | Method | Training-free | PickCan | MoveNear | Drawer | DrawerApple | Average | FLOPs↓ | Speedup↑ | Params (B) |
|---|---|---|---|---|---|---|---|---|---|---|
| | CogACT | - | 91.3% | 85.0% | 71.8% | 50.9% | 74.8% | 100.0% | 1.00× | 7.63 |
| | Random Dropping | ✓ | 9.7% | 20.4% | 53.5% | 0.0% | 20.9% | 58.5% | 1.20× | 7.63 |
| Visual | FastV | ✓ | 92.6% | 81.4% | 69.8% | 52.4% | 74.1% | 42.0% | 1.21× | 7.63 |
| Matching | VLA-Cache | ✓ | 92.0% | 83.3% | 70.5% | 51.6% | 74.4% | 80.1% | 1.38× | 7.63 |
| | EfficientVLA (L=28, T=112) | ✓ | 95.3% | 83.3% | 70.3% | 56.5% | 76.4% | 45.1% | 1.59× | 5.87 |
| | EfficientVLA (L=28, T=56) | ✓ | 94.7% | 82.4% | 69.8% | 55.4% | 75.5% | 32.9% | 1.71× | 5.87 |
| | EfficientVLA (L=22, T=112) | ✓ | 94.0% | 82.1% | 69.2% | 54.6% | 75.0% | 38.2% | 1.78× | 4.86 |
| | EfficientVLA (L=22, T=56) | ✓ | 93.3% | 81.3% | 68.2% | 53.8% | 74.2% | 28.9% | 1.93× | 4.86 |
| | CogACT | - | 89.6% | 80.8% | 28.3% | 46.6% | 61.3% | 100.0% | 1.00× | 7.63 |
| | Random Dropping | ✓ | 4.0% | 16.1% | 15.6% | 0.0% | 8.9% | 58.5% | 1.20× | 7.63 |
| Variant | FastV | ✓ | 91.4% | 78.6% | 27.6% | 50.6% | 62.1% | 42.0% | 1.19× | 7.63 |
| Aggregation | VLA-Cache | ✓ | 91.7% | 79.3% | 32.5% | 45.8% | 62.3% | 82.6% | 1.37× | 7.63 |
| | EfficientVLA(L=28, T=112) | ✓ | 94.8% | 77.6% | 28.4% | 51.9% | 63.2% | 45.1% | 1.57× | 5.87 |
| | EfficientVLA (L=28, T=56) | ✓ | 94.4% | 77.2% | 27.6% | 51.3% | 62.6% | 32.9% | 1.69× | 5.87 |
| | EfficientVLA (L=22, T=112) | ✓ | 93.9% | 76.4% | 27.3% | 50.6% | 62.1% | 38.2% | 1.76× | 4.86 |
| | EfficientVLA (L=22, T=56) | ✓ | 93.2% | 75.8% | 26.9% | 49.2% | 61.2% | 28.9% | 1.91× | 4.86 |

### 3.4.2 Static N-Step Caching Implementation

We define a cache interval $N$ ($1 \leq N < T_{start}$). At the initial timestep $t = T_{start}$, the features $\mathbf{h}^{\text{attn}}_{T_{start}}$ and $\mathbf{h}^{\text{mlp}}_{T_{start}}$ are computed via Equations 6 and 7 and stored in a persistent cache, denoted $\mathcal{C}_{attn}$ and $\mathcal{C}_{mlp}$. For any subsequent timestep $t < T_{start}$, these features are recomputed and the caches are updated if and only if $t \pmod{N} = 0$ (assuming $t > 0$ and $t$ aligns with desired multiples for caching, e.g., $T_{start}, T_{start} - N, T_{start} - 2N, \dots$). Thus, for such recomputation timesteps:

$$\mathcal{C}_{attn} \leftarrow \text{Self-Attn}(\mathbf{z}_t) \qquad (8)$$

$$\mathcal{C}_{mlp} \leftarrow \text{MLP}(\mathcal{C}_{attn} + \mathbf{z}_t) \qquad (9)$$

And the outputs for this step are $\mathbf{h}^{\text{attn}}_t = \mathcal{C}_{attn}$ and $\mathbf{h}^{\text{mlp}}_t = \mathcal{C}_{mlp}$. In all other timesteps, where $t \pmod{N} \neq 0$, the computationally intensive Self-Attn and MLP operations are entirely bypassed. Instead, the required features are directly retrieved from the most recently populated cache:

$$\mathbf{h}^{\text{attn}}_t \leftarrow \mathcal{C}_{attn} \quad (\text{when } t \pmod{N} \neq 0) \qquad (10)$$

$$\mathbf{h}^{\text{mlp}}_t \leftarrow \mathcal{C}_{mlp} \quad (\text{when } t \pmod{N} \neq 0) \qquad (11)$$

This static caching schedule effectively prunes the execution of these core modules for $N - 1$ out of every $N$ timesteps post-initialization, leading to a substantial reduction in floating-point operations and latency for the action generation component of the VLA. The choice of $N$ allows for a tunable trade-off between acceleration and the fidelity of the generated actions, as reusing features for longer intervals might introduce slight deviations if underlying representations were to change rapidly.

## 4 Experiment

### 4.1 Experimental Settings

**Simulation Implementation Details**. To assess our VLA model, we utilize the SIMPLER environment [21], a simulation-based benchmark for table-top manipulation. SIMPLER is designed to closely emulate real-world dynamics for robots such as the Google Robot and WidowX, demonstrating robust alignment between simulation and real-world performance. The VLA model in this setup takes 224×224 RGB image observations and natural language task instructions (e.g., "Pick coke can") as input and outputs a sequence of actions in 7-DoF Cartesian space. The SIMPLER supports two evaluation configurations: *Visual Matching*, which prioritizes fidelity to real-world appearances, and *Variant Aggregations*, which incorporates diverse conditions such as altered lighting, backgrounds, and surface textures. For the Google robot, SIMPLER provides both two evaluation settings, each featuring the same four tasks: 1) Pick coke can; 2) Move near; 3) Open/close drawer; and 4) Open top drawer and place apple. Success rate is used as the evaluation metric.

**Baselines**. Our primary experimental validation of EfficientVLA is performed on the *CogACT* [41], which integrates powerful vision encoders (DINOv2 [39] and SigLIP [40]), a Llama2-7B [14]

Table 3: **Scalability Analysis**: We evaluate mean success rate and inference time in a simulation environment of visual matching across various model sizes. Our EfficientVLA configuration maintains L = 22 and T = 56.

| SIMPLER | Model | Action-Params | Methods | Average | Inference time (s) | Total-Params (B) |
|---------|-------|---------------|---------|---------|--------------------|-----------------| 
| Visual | CogACT-Small | 13M | CogACT | 73.3% | 0.2156 | 7.55 |
| | | | EfficientVLA | 72.6% | 0.1173 | 4.78 |
| Matching | CogACT-Base | 89M | CogACT | 74.8% | 0.2342 | 7.63 |
| | | | EfficientVLA | 74.2% | 0.1213 | 4.86 |
| | CogACT-Large | 308M | CogACT | 76.7% | 0.2628 | 7.85 |
| | | | EfficientVLA | 76.1% | 0.1312 | 5.08 |

language module for multimodal reasoning, and a Diffusion Transformer (DiT) for generating action trajectories. We benchmark against relevant baseline methodologies. These include a *Random Dropping* approach, where 112 visual tokens are retained uniformly at random, to evaluate the benefits of our guided vision token pruning. We further compare with *FastV* [42], a notable approach focused on accelerating inference by pruning redundant visual tokens, and *VLA-Cache* [20], which leverages temporal analysis to cache static tokens across timesteps. The experimental results and model details for $\pi_0$ on the LIBERO [43] can be found in the Appendix.

**Implementation Details**. For EfficientVLA, in addition to layer pruning, we further compressed the model parameters by adopting the PruneNet [44] configuration for LLM compression. Specifically, we applied a sparsity of 25% to the MLP layers of all Transformer blocks. For visual token pruning, we started from the 2nd Transformer layer with a ratio $\alpha = 50\%$ and $K_{key} = 4$ for key task-critical tokens. Furthermore, the cache interval was set to 5. All experiments were conducted on NVIDIA A40 GPUs, and the inference time was measured as the average single-step inference duration. More details can be found in the supplementary material.

## 4.2 Results on Simulation Environment

**Main Results on SIMPLER**. Table 2 details the performance of our structured, entirely training-free pruning method in the SIMPLER environment. Our approach consistently excels across configurations retaining 22/28 layers and 56/112 visual tokens. For instance, pruning 10 layers with 112 tokens surpasses both CogACT and VLA-cache in success rate and inference speed. Remarkably, on the *pick coke can* task, pruning 36% of parameters paradoxically improved the success rate from 91.3% to 94.0%, highlighting significant parameter redundancy in the VLA model. Conversely, random token dropping to 112 tokens drastically reduces the average success rate to 20.9%, affirming the superiority of our guided selection strategy. Furthermore, a 22-layer, 56-token setup achieved a 71.1% reduction in FLOPs with merely a 0.6% drop in average success rate, demonstrating exceptional efficiency. In comparision, approaches like FastV [42] (T = 56) show that solely optimizing visual tokens yields only a 1.21× speedup due to unaddressed memory bottlenecks, despite acceptable task performance.

**Efficiency Analysis**. As demonstrated in Table 2 and Figure 3, our proposed method significantly outperforms previous baselines, achieving a 71.1% reduction in FLOPs and a 1.93× speedup in inference time. In stark contrast, VLA-cache, when applied to the CogACT model, only reduces FLOPs by 19.9% and achieves a mere 1.38× speedup. This disparity substantiates our prior analysis: VLA-cache, functioning solely as a cache for visual tokens between adjacent time steps, is inherently constrained by memory bounds, thereby limiting the efficacy of token-only acceleration. Consequently, the structured framework of our system offers distinct advantages, highlighting our method's superior capability in balancing computational efficiency with robust performance.

**Scalability Evaluation**. Table 3 illustrates the scalability of our proposed method across different sizes of the CogACT model. With the primary difference among the three models being the parameter size of their action modules, the results reveal that our method's effectiveness becomes more pronounced with larger models. Specifically, on the CogACT-Large model, our approach achieves a 2.0× inference speedup, while performance only marginally decreases from 76.7% to 76.1%. This increased impact is because action modules in larger models, having more parameters, inherently

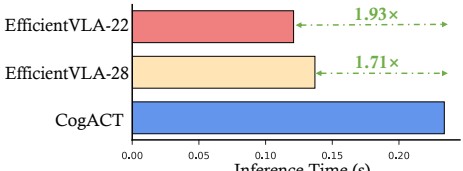 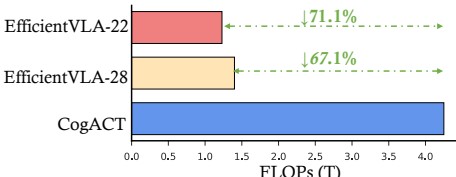

Figure 3: **Efficiency analysis** in simulation, comparing FLOPs and inference time of our EfficientVLA variants against the original model backbone. EfficientVLA-22 and EfficientVLA-28 denote configurations retaining 22 and 28 LLM layers, respectively.

Table 4: Performance impact of varying visual token reduction ratios (Left) and diffusion action head cache intervals (Right) as applied within our EfficientVLA framework.

| Token Ratio | 56 77.8% | 72 71.8% | 96 62.5% | 112 56.2% | 256 100.0% |
|---|---|---|---|---|---|
| Accuracy | 95.0% | 95.3% | 95.0% | 96.0% | 91.3% |
| Inference time (s) | 0.1866 | 0.1870 | 0.1889 | 0.1956 | 0.2342 |
| FLOPs (T) | 1.76 | 1.96 | 2.25 | 2.45 | 4.19 |

| Cache Interval | 1 | 2 | 3 | 4 | 5 |
|---|---|---|---|---|---|
| Accuracy | 91.3% | 94.0% | 93.7% | 90.3% | 93.7% |
| Inference time (s) | 0.2342 | 0.2031 | 0.1987 | 0.1953 | 0.1909 |
| FLOPs (T) | 4.190 | 4.161 | 4.155 | 4.150 | 4.144 |

Table 5: **Ablation study** on our EfficientVLA, where 'Layer' denotes applying only the LLM layer pruning component, and 'MLP' refers to a distinct strategy of compressing 25% of MLP weights within each layer.

| | Model Compression | | Visual Token | Action | Success | Inference | Speedup↑ |
|---|---|---|---|---|---|---|---|
| | **Layer** | **MLP** | **Pruning** | **Cache** | **Rate** | **Time (s)** | |
| Ex0 | ✗ | ✗ | ✗ | ✗ | 91.3% | 0.2342 | 1.00× |
| Ex1 | ✗ | ✗ | ✓ | ✗ | 95.6% | 0.1866 | 1.25× |
| Ex2 | ✗ | ✗ | ✗ | ✓ | 93.7% | 0.1909 | 1.23× |
| Ex3 | ✓ | ✗ | ✓ | ✗ | 85.7% | 0.1604 | 1.46× |
| Ex4 | ✓ | ✓ | ✗ | ✗ | 92.3% | 0.1638 | 1.43× |
| Ex5 | ✓ | ✓ | ✗ | ✓ | 93.3% | 0.1387 | 1.69× |
| Ex6 | ✗ | ✗ | ✓ | ✓ | **95.3%** | 0.1592 | 1.47× |
| Ex7 | ✓ | ✓ | ✓ | ✓ | 93.3% | **0.1213** | **1.93×** |

exhibit longer inference times, thus allowing our method to yield more significant accelerations. These findings also underscore the robustness of our method across models of various scales.

**Impact of Token Reduction Ratio and Cache Interval.** Table 4 details our experiments on the *pick coke can* task reveal distinct impacts of component-specific optimizations. Aggressively pruning visual tokens, down to retaining only 22%, effectively lessens computational load for substantial, near-lossless acceleration. However, inference speed gains largely saturate beyond this point, with further token reduction yielding only marginal improvements, thereby revealing dominant system-level performance bottlenecks. Separately, for the diffusion-based action generator, we observed that increasing the cache reuse interval $N$ for intermediate attention and MLP features progressively and significantly accelerates action trajectory generation.

## 4.3 Ablation Study

We conducted an ablation study on the components of our proposed framework, using the *pick coke can* task as an illustrative example. As suggested by prior analysis (e.g., Figure 1 (a), solely optimizing visual tokens for VLA inference tasks yields limited acceleration; retaining just 56 tokens resulted in a mere 1.23× speedup, although the success rate paradoxically rose from 91.3% to 95.6%. This underscores the inherent limitations of token-centric optimization methods, such as VLA-cache, and affirms that achieving substantial VLA inference acceleration viable for hardware deployment necessitates a more model-centric strategy. In contrast, our model compression approach—concurrently pruning layers and compressing MLP in the remaining layers—achieved a 1.43× speedup. Critically, when all components were integrated, a 1.93× speedup was realized, and the overall task success rate still saw an improvement of 2% points. These collective results highlight the imperative and significance of adopting a structured framework for effective VLA inference acceleration.

# 5  Conclusion

In this paper, we addressed the critical challenge of high computational and memory overheads that impede the practical deployment of powerful Diffusion-based Vision-Language-Action (VLA) models. We proposed EfficientVLA, a novel training-free, structured framework to accelerate VLA models. Our framework enhances efficiency by synergistically pruning redundant layers of language module identified by their minimal impact on transforming hidden states and by strategically selecting a compact set of visual tokens that balances VLA task relevance with inherent feature diversity. Furthermore, it optimizes the action module by caching critical intermediate computations across its iterative denoising steps. We demonstrate the efficacy of EfficientVLA through extensive experiments on the CogACT in the SIMPLER environment [21], achieving a $1.93\times$ inference speedup and reducing FLOPs to $28.9\%$, all while incurring a minimal accuracy degradation of only $0.6\%$.

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

# A  Experimental Settings

## A.1  SIMPLER Environment

The SIMPLER simulation environment serves as our primary benchmark for evaluating CogACT models. It is specifically designed to closely mirror real-world robotic setups, thereby facilitating more realistic evaluations and bridging the real-to-sim control and visual gap.

SIMPLER offers two distinct evaluation settings:

- **Visual Matching (VM):** This setting closely replicates real-world tasks by minimizing discrepancies between the simulated and real environments, prioritizing fidelity to real-world appearances.
- **Variant Aggregation (VA):** This setting builds upon Visual Matching by introducing variations to elements such as background, lighting, distractors, and table texture, challenging models to generalize across diverse conditions.

For the Google robot setup, SIMPLER provides both evaluation settings, each featuring the same four tasks: 1) Pick coke can, 2) Move near, 3) Open/close drawer, and 4) Open top drawer and place apple. These four tasks for the Google robot are also illustrated in Figure 4.

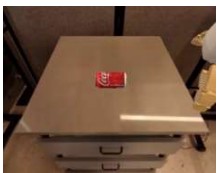 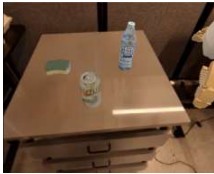 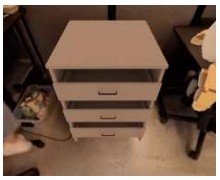 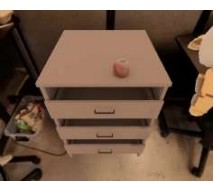

(a) Pick coke can     (b) Move near     (c) Open/close drawer     (d) Open top drawer and place apple

Figure 4: Representative robotic manipulation tasks for the Google robot in the SIMPLER environment: (a) Pick coke can, (b) Move near, (c) Open/close drawer, and (d) Open top drawer and place apple.

## A.2  Baselines

We benchmark our EfficientVLA against the following relevant baseline and backbone methodologies:

- **CogACT:** This model serves as our primary experimental validation platform. CogACT integrates powerful vision encoders (DINOv2 and SigLIP) to process raw visual input, a Llama2-7B language module for multimodal reasoning, and a Diffusion Transformer (DiT) as a specialized action module for generating precise action trajectories. It aims to synergize "cognition" (VLM output) with "action" capabilities by conditioning the diffusion-based action module on the VLM's extracted features, addressing the continuous, multimodal, and temporally correlated nature of robot actions.
- $\pi_0$**:** This model is a prototype generalist robot policy developed by prior research to address the versatility gap between human and machine intelligence. Built upon a pre-trained vision-language model to harness Internet-scale image-text corpora for semantic reasoning and problem-solving, $\pi_0$ is extended into a VLA model through cross-embodiment training, integrating diverse data from various robot types to enhance generalization and mitigate data scarcity. Featuring an action chunking architecture with flow matching, it enables high-frequency control up to 50 Hz for dexterous tasks such as laundry folding, and is pre-trained on over 10,000 hours of robot data before fine-tuning on curated datasets to optimize dexterity, efficiency, and robustness, serving as a robust reference for evaluating our proposed EfficientVLA framework.
- **VLA-Cache:** This is a training-free acceleration method designed to improve VLA model efficiency in robotic manipulation. VLA-Cache operates on the principle that visual inputs in

sequential robotic tasks often exhibit minimal variation between successive steps, particularly in background regions. It incorporates a token-selection mechanism that identifies visually static tokens with minimal changes from the previous step and reuses their computational results via KV-cache. Additionally, it includes a fine-grained selection scheme to filter out critical task-relevant tokens, ensuring they undergo full computation to preserve accuracy, and employs a layer-adaptive strategy to adjust reuse ratios based on attention concentration.

- **FastV:** This approach focuses on accelerating inference in Large Vision-Language Models (LVLMs) by pruning redundant visual tokens. FastV's core insight is the identification of inefficient attention phenomena in deeper layers of popular LVLMs, where image tokens receive significantly less attention despite accounting for a large portion of input tokens. It proposes a plug-and-play method that dynamically prunes a percentage of these less impactful visual tokens after a specific layer, guided by attention scores, to reduce computational costs (FLOPs) without sacrificing performance.

# B  More Experimental Results

## B.1  Results on Vision-Language Tasks

Table 6 illustrates the performance of our EfficientVLA token pruning strategy in comparison to the vanilla LLaVA-1.5-7B baseline and the conventional FastV method across a suite of multimodal vision-language benchmarks. Our approach, which employs an advanced token pruning mechanism tailored to preserve semantically critical visual tokens while aggressively reducing redundancy, consistently outperforms FastV at equivalent token retention levels. Specifically, at 192 tokens, EfficientVLA achieves an average performance of 98.0%, retaining nearly all baseline efficacy, whereas FastV drops to 91.2%; similarly, at 128 tokens, our method yields 96.6% average accuracy, markedly superior to FastV's 86.4%. This superior retention of multimodal understanding capabilities underscores EfficientVLA's efficacy in mitigating information loss during pruning, enabling substantial inference acceleration in vision-language tasks without compromising downstream task performance.

Table 6: Performance evaluation of the EfficientVLA token pruning strategy compared to the vanilla LLaVA-1.5-7B baseline and the traditional FastV method across diverse multimodal vision-language benchmarks.

| Method | Benchmarks | | | | | | | | | | Avg |
| --- | --- | --- | --- | --- | --- | --- | --- | --- | --- | --- | --- |
| | GQA | MMB | MMB-CN | MME | POPE | SQA | VQA-v2 | VQA-Text | VizWiz | OCRBench | |
| LLaVA-1.5-7B (Vanilla, 576 tokens) | 61.9 | 64.7 | 58.1 | 1862 | 85.9 | 69.5 | 78.5 | 58.2 | 50.0 | 297 | 100.0% |
| +FastV (192 tokens) | 52.7 | 61.2 | 57.0 | 1612 | 64.8 | 67.3 | 67.1 | 52.5 | 50.8 | 291 | 91.2% |
| +Ours (192 tokens) | 58.5 | 62.3 | 57.0 | 1823 | 82.8 | 69.6 | 76.7 | 57.6 | 51.0 | 292 | 98.0% |
| +FastV (128 tokens) | 49.4 | 56.1 | 56.4 | 1490 | 56.0 | 60.2 | 61.8 | 50.6 | 51.3 | 285 | 86.4% |
| +Ours (128 tokens) | 58.2 | 61.5 | 56.6 | 1763 | 79.8 | 68.5 | 75.7 | 56.1 | 51.4 | 289 | 96.6% |

## B.2  More Generalizability Experiments

To validate the generalizability of our proposed method, we conducted additional experiments on the model using the LIBERO [43] benchmark, fine-tuning it for 30k steps as a baseline and performing inference on a NVIDIA 4090. The LIBERO benchmark evaluates lifelong robotic manipulation in four challenging task suites, LIBERO-Spatial, LIBERO-Object, LIBERO-Goal and LIBERO-10, introducing variations in spatial layouts, object selection, task goals, and long-horizon planning, which test a model's ability to generalize across diverse manipulation scenarios. As shown in Table 7, our training-free approach was evaluated on these four tasks. Random Dropping, which randomly prunes 112 visual tokens, achieved a success rate of only 18.5% with limited acceleration (1.14×). FastV provided a modest 1.16× speedup but incurred performance loss. In contrast, EfficientVLA without pruning already achieved a 1.46× speedup with only a 1.8% performance drop. With pruning, EfficientVLA delivered a 1.71× speedup, increasing inference frequency from 11.65 Hz to 19.96 Hz, while outperforming FastV in success rate. These results demonstrate the method's robustness and scalability across diverse VLA models and datasets.

Table 7: Quantitative assessment of performance and efficiency metrics for the $\pi_0$ baseline model and comparative methods on the LIBERO benchmark, evaluating task-specific success rates (Spatial, Object, Goal, 10), average accuracy, latency, speedup, and operational frequency.

| Method | Spatial | Object | Goal | 10 | Avg. | Latency/s | Speedup | Freq. (Hz) |
|---|---|---|---|---|---|---|---|---|
| $\pi_0$ | 98.2 | 98.2 | 94 | 83.2 | 93.4 | 0.0858 | - | 11.65 |
| Random Dropping | 36.8 | 21.4 | 15.8 | 0 | 18.5 | 0.0753 | 1.14× | 13.28 |
| FastV | 95.2 | 96.6 | 90.2 | 79.2 | 90.3 | 0.0739 | 1.16× | 13.53 |
| **EfficientVLA w/o prune** | 95.8 | 97.8 | 91.6 | 81.2 | 91.6 | 0.0588 | 1.46× | 17.01 |
| **EfficientVLA** | 95.2 | 97.2 | 90.6 | 78.6 | 90.4 | 0.0501 | 1.71× | 19.96 |

## C   Related Work on Inference Acceleration Techniques

A variety of training-free methods have emerged to expedite inference in large-scale models by leveraging layer pruning, token pruning, and temporal caching mechanisms, thereby reducing computational overhead without necessitating model retraining. In the context of layer pruning for large language models (LLMs), Yuan et al. [45] mitigate redundancy by collapsing multiple redundant layers into a single effective layer, preserving essential functionalities; SlimGPT [46] applies batched greedy pruning coupled with incremental pruning ratios to systematically minimize error accumulation during compression; and TrimLLM [47] enables progressive and targeted layer dropping by exploiting layer-wise specialization, allowing for adaptive model thinning tailored to specific tasks. For visual token pruning in large vision-language models (LVLMs), FastV [42] executes dynamic token elimination guided by real-time attention scores to streamline processing; SparseVLM [48] introduces a text-guided paradigm for rank-based pruning, further enhanced by token recycling to retain informative elements; PruMerge [49] adaptively merges redundant tokens by capitalizing on inherent attention sparsity patterns; and MustDrop [50] refines token selection and retention strategies across distinct phases including encoding, prefilling, and decoding, ensuring minimal performance degradation. Regarding temporal caching in diffusion transformers, FORA [51] supports seamless direct reuse of cached intermediate features in a training-free manner, accelerating iterative generation processes; while ToCa [52] employs token-wise caching with dynamically adaptive retention ratios, optimizing memory usage and inference speed across varying sequence lengths.

## D   Impact Statement

This paper introduces EfficientVLA, a crucial contribution to Vision-Language-Action (VLA) models by directly addressing their significant computational and memory demands through a novel training-free framework. EfficientVLA synergistically prunes redundant language layers, optimizes visual token selection for task-relevance and diversity, and caches intermediate features in the diffusion-based action head, thereby significantly boosting VLA model efficiency and speed. This enables the practical deployment of powerful VLA models on resource-constrained robotic platforms for real-time interaction, accelerating progress in robotic manipulation and reasoning tasks, and contributing to more development by reducing computational load. We intend this work for ethical academic research and authorized commercial applications and strictly prohibit its use for harmful, unethical, or unlawful robotic actions, underscoring our responsibility to ensure societal benefit aligned with fundamental ethical principles.

## E   Limitations

While EfficientVLA significantly advances VLA model acceleration, certain limitations warrant discussion. Our training-free approach, may not achieve the maximal compression or speedup attainable by training-aware methods. The fixed cache interval $N$ in the action head introduces a trade-off between acceleration and action fidelity and future work could explore adaptive caching. Furthermore, due to the limited availability of open-source diffusion-based VLA models, our current demonstrations are primarily on CogACT. In the future, we aim to validate scalability and effective-

ness of EfficientVLA across a wider range of models and tasks as more such architectures become available.

