# OpenReview forum: "EfficientVLA: Training-Free Acceleration and Compression for Vision-Language-Action Models"
_NeurIPS.cc/2025/Conference — NeurIPS 2025 poster_

### Official Review · Reviewer_nRSS · 2025-07-01

**Clarity:** 3
**Significance:** 3
**Originality:** 3
**Rating:** 5
**Confidence:** 3

**Summary:**

This paper aims to improve the inference efficiency of pretrained VLA models and introduces a training-free framework named EfficientVLA. The proposed efficiency improvements are achieved through three key techniques: selective vision token pruning, LLM layer pruning, and action head computation caching. Compared to the vanilla model, EfficientVLA achieves a 1.93× inference speedup under the SIMPLER setting, with only a 0.6% drop in performance.

**Questions:**

1. Based on Table 5, visual token pruning appears to improve visual success rate, whereas model compression and action cache mainly reduce computational costs. Could the authors elaborate on this finding and provide further discussion?
2. Regarding the action head caching, could the authors clarify how feature updates are handled when self-attention and MLP layers are bypassed? An explicit explanation of the feature flow in this case would greatly help reader comprehension.

**Ethical Concerns:**

["NO or VERY MINOR ethics concerns only"]

**Final Justification:**

I thank the authors for their detailed response, which has clearly addressed my concerns. Given the paper’s broad relevance to the VLA community, I am raising my score to accept.

**Limitations:**

While the technical novelty of the proposed methods is limited, since similar techniques have been explored in the VLM or LLM domains, their integration into the VLA context is also important. The paper effectively adapts existing ideas to a new domain, offering meaningful efficiency improvements.

**Paper Formatting Concerns:**

No formatting concerns.

**Quality:**

3

**Strengths And Weaknesses:**

**Strengths**
1. The redundancy analysis presented in Figure 1 provides valuable insight into the computational inefficiency in VLA models.
2. The proposed framework achieves state-of-the-art inference efficiency with minimal performance degradation, demonstrating the effectiveness of combining the three techniques.

**Weaknesses**
1. The framework is only evaluated on the CogACT model. Experiments on additional VLA architectures would strengthen the generalizability and robustness of the method.
2. More ablations on visual token reduction are needed. For instance, analyzing the individual contributions of Vkey, Vtask, and Vdiv would clarify their roles. Additionally, visualizations of the token pruning process would improve interpretability.

**Minor**
1. It seems that $r_i$ in Equation (2) and $s_i$ in Section 3.3.2 refer to the same variable. Please clarify or correct if this is a typo.
2. In Table 4, is the ratio shown the token reduction ratio or the token preservation ratio? Also, why is the last column showing a 100% ratio while its FLOPs are the highest among all columns?

---

> ### Author Rebuttal · Authors · 2025-07-31
>
> We extend our heartfelt thanks to the reviewer for their thorough and constructive input, which has been instrumental in shaping this work.
>
> **Weaknesses**
>
> ***W1:** The framework is only evaluated on the CogACT model. Experiments on additional VLA architectures would strengthen the generalizability and robustness of the method.*
>
> **A1:** Thank you for your valuable suggestion. To validate the generalizability of our proposed method, we conducted additional experiments on the $\pi_0$ model using the LIBERO benchmark, fine-tuning it for 30k steps as a baseline and performing inference on a NVIDIA 4090. The LIBERO benchmark evaluates lifelong robotic manipulation across four challenging task suites—LIBERO Spatial, LIBERO-Object, LIBERO-Goal, and LIBERO-10—introducing variations in spatial layouts, object selection, task goals, and long-horizon planning, which test a model’s ability to generalize across diverse manipulation scenarios.
>
> As shown in Table 1, our training-free approach was evaluated on these four tasks. Random Dropping, which randomly prunes 112 visual tokens, achieved a success rate of only 18.5% with limited acceleration (1.14X). FastV provided a modest 1.16X speedup but incurred performance loss. In contrast, EfficientVLA without pruning already achieved a 1.46X speedup with only a 1.8% performance drop. With pruning, EfficientVLA delivered a 1.71X speedup, increasing inference frequency from 11.65 Hz to 19.96 Hz, while outperforming FastV in success rate. These results demonstrate the method’s robustness and scalability across diverse VLA models and datasets. Addressing the SIMPLER benchmark’s limited diversity, LIBERO’s broader scope strengthens our evaluation, and further experiments on additional benchmarks are planned for future work. We appreciate your suggestion, which has helped make our work more complete.
>
> **Table 1:** Performance and efficiency metrics of EfficientVLA and baselines on LIBERO benchmark tasks.
>
> | LIBERO                 | Spatial | Object | Goal | 10   | Avg. | Latency/s↓ | Speedup↑ | Frequency↑ (Hz) |
> | ---------------------- | ------- | ------ | ---- | ---- | ---- | ---------- | -------- | --------------- |
> | $\pi_0$                | 98.2    | 98.2   | 94   | 83.2 | 93.4 | 0.0858     | -        | 11.65           |
> | Random Dropping        | 36.8    | 21.4   | 15.8 | 0    | 18.5 | 0.0753     | 1.14X    | 13.28           |
> | FastV                  | 95.2    | 96.6   | 90.2 | 79.2 | 90.3 | 0.0739     | 1.16X    | 13.53           |
> | EfficientVLA w/o prune | 95.8    | 97.8   | 91.6 | 81.2 | 91.6 | 0.0588     | 1.46X    | 17.01           |
> | EfficientVLA           | 95.2    | 97.2   | 90.6 | 78.6 | 90.4 | 0.0501     | 1.71X    | 19.96           |
>
>
>
> ***W2:** More ablations on visual token reduction are needed. For instance, analyzing the individual contributions of Vkey, Vtask, and Vdiv would clarify their roles. Additionally, visualizations of the token pruning process would improve interpretability.*
>
> **A2:** Thank you for your valuable suggestions. We appreciate the call to analyze the individual contributions of Vkey, Vtask, and Vdiv to clarify their roles.
>
> - Vkey is selected as the top $K_ {key}$ tokens (empirically set between 4 and 8) based on normalized task relevance scores $s_i$, derived from cross-attention scores  $A^{(h)}_ {i,j}$ averaged across H heads and summed over $L_ {ctx}$ contextual elements. This set forms the core visual cues essential for task understanding and execution, providing a foundational scaffold for $V_ {pruned}$.
>
> - Vtask, selecting $K_ {task} = \lfloor \alpha \cdot K_ {aug} \rfloor$ high-relevance tokens from the remaining $V_{rem}$, enhances task-oriented semantics by refining details that Vkey might overlook. This step leverages the hybrid selection strategy guided by the parameter $\alpha$, ensuring alignment with language instructions and improving task-specific performance.
>
> - Vdiv, choosing $K_ {div} = K_ {aug} - K_ {task}$ tokens based on cosine distance, boosts diversity and robustness to complex or unseen scenarios by incorporating distinct visual features. This diversity-driven augmentation, critical for generalization across varied environments, mitigates overfitting to core tokens. We will expand on these contributions with further analysis in the revised manuscript.
>
> We conducted further ablations on "PickCan" with token pruning only, retaining 56 tokens. Table 2 explores Vkey’s impact with Vtask:Vdiv = 1:1, showing suboptimal success rates (SR) when Vkey is overly narrow or excessive, with an optimal range around 4-8 tokens. In Table 3, with Vkey = 4, we varied Vtask:Vdiv ratios, revealing a need to balance task relevance and diversity; a low Vdiv ratio risks overfitting, while balanced ratiosoptimize performance.
>
> **Table 2:** Ablation results for Vkey impact on "PickCan" success rate (SR) with 56 tokens and Vtask:Vdiv = 1:1.
>
> | Vkey  | 1    | 2    | 4    | 8    | 12   |
> | ----- | ---- | ---- | ---- | ---- | ---- |
> | SR(%) | 89.6 | 94.2 | 95   | 93.3 | 91.3 |
>
> **Table 3:** Ablation results for Vtask:Vdiv ratio impact on "PickCan" success rate (SR) with 56 tokens and Vkey = 4.
>
> | Vtask：Vdiv | 5 ：1 | 2 ：1 | 1：1 | 1： 2 | 1 ：5 |
> | ----------- | ----- | ----- | ---- | ----- | ----- |
> | SR(%)       | 88    | 91.3  | 95   | 94.6  | 94.2  |
>
> Due to rebuttal policy constraints, we cannot provide visualizations at this stage, but we commit to including pruning process visualizations in the revised manuscript to enhance interpretability, and we appreciate the suggestion.
>
>
>
> **Minor**
>
> ***M1:** It seems  that $r_i$ in Equation (2) and $s_i$ in Section 3.3.2 refer to the same variable. Please clarify or correct if this is a typo.*
>
> **A1:** Thank you for your comment. We clarify that $r_i$ in Equation (2) and $s_i$ in Section 3.3.2 are distinct variables. $r_i$ represents the raw task relevance score for each visual token $v_i$, computed by averaging attention scores $A^{(h)}_ {i,j}$ across H heads and summing over $L_ {ctx}$ contextual elements. $s_i$, however, is the normalized version of $r_i$, used for robust comparison and selection of the $K_ {key}$ tokens in $V_ {key}$. This normalization step ensures consistency in token pruning. We will enhance the notation clarity in the revised manuscript.
>
> ***M2:** In Table 4, is the ratio shown the token reduction ratio or the token preservation ratio? Also, why is the last column showing a 100% ratio while its FLOPs are the highest among all columns?*
>
> **A2:** Thank you for your comments. In Table 4, the ratio is the token pruning ratio, not preservation; for 256 tokens, it should be 0. We apologize for the oversight, and thank you for pointing this out! We will clarify this in the updated table and accompanying text.
>
>
>
> **Questions:**
>
> ***Q1:** Based on Table 5, visual token pruning appears to improve visual success rate, whereas model compression and action cache mainly reduce computational costs. Could the authors elaborate on this finding and provide further discussion?*
>
> **A1:** Thank you for the reviewer’s insightful comment. Based on Table 5, we elaborate on the findings regarding visual token pruning, model compression, and action caching. Visual token pruning appears to enhance visual success rates for certain tasks, such as "PickCan," likely due to implicit regularization that removes redundant information, thereby improving performance. This underscores the widespread presence and detrimental impact of redundancy in VLA tasks. However, for some tasks like "MoveNear," pruning 77.8% of tokens reduces performance from 80.8% to 78.6%, which we attribute to task-specific characteristics influencing the necessity of certain visual cues.
>
> In contrast, model compression and action caching primarily reduce computational costs, achieving a 1.69× acceleration. We consider this a key bottleneck in accelerating Diffusion-based VLA models. Nevertheless, visual token pruning remains essential, as it significantly lowers the overall computational load, reducing FLOPs from 100% to 42%, predominantly driven by the LLM module. These insights will be further detailed in the revised manuscript.
>
> ***Q2:** Regarding the action head caching, could the authors clarify how feature updates are handled when self-attention and MLP layers are bypassed? An explicit explanation of the feature flow in this case would greatly help reader comprehension.*
>
> **A2:** Thank you for your suggestion. Regarding action head caching, we clarify the feature update process when bypassing self-attention and MLP layers in the diffusion cache. Our approach, detailed in Section 3.4, leverages the temporal coherence of intermediate features (e.g., $ h^{attn}_ t $ and $ h^{mlp}_ t $ from Equations 6 and 7) observed in the Diffusion Transformer  blocks (Figure 1(c)). At the initial timestep $ t = T_{start} $, we compute these features fully using self-attention and MLP on input $ z_t $, caching them as $ C^{attn} $ and $ C^{mlp} $. For subsequent timesteps, we recompute and update the caches at fixed intervals N (e.g., when $ t \mod N = 0 $, per Equations 8 and 9), reusing the cached values (Equations 10 and 11) for intervening steps to bypass these computations.
>
> This static N step caching reduces redundant calculations across T timesteps, achieving significant computational savings. Feature updates occur periodically to refresh outdated representations, balancing efficiency and action fidelity. We will include this detailed feature flow explanation in the revised manuscript to enhance comprehension.

---

> > ### Comment · Reviewer_nRSS · 2025-08-04
> >
> > I thank the authors for their detailed response, which has clearly addressed my concerns. Given the paper’s broad relevance to the VLA community, I am raising my score to accept.

---

### Official Review · Reviewer_9KWi · 2025-07-03

**Clarity:** 3
**Significance:** 3
**Originality:** 2
**Rating:** 4
**Confidence:** 2

**Summary:**

This paper presents EfficientVLA, a training-free, structured inference acceleration framework for diffusion-based Vision-Language-Action (VLA) models. The method systematically addresses inefficiencies across the VLA pipeline via three key components: (1) non-contiguous pruning of redundant layers in the language module based on inter-layer similarity; (2) task-aware and diversity-promoting visual token pruning; and (3) temporal feature caching within the diffusion-based action head. Experiments on the CogACT model in the SIMPLER environment show up to 1.93× inference speedup and 71.1% FLOPs reduction with negligible accuracy loss (<0.6%).

**Questions:**

- Do redundancies in different modules interact with each other? For example, does pruning in the language module influence token selection or the effectiveness of diffusion caching? Is there a trade-off involved when reducing redundancy across different stages of the VLA pipeline?

- Can EfficientVLA generalize to other multimodal tasks beyond robotics? In particular, could the techniques for visual token pruning and temporal caching be applied to VQA, image captioning, or other vision-language domains?

**Ethical Concerns:**

["NO or VERY MINOR ethics concerns only"]

**Final Justification:**

The authors have addressed some of my concerns. However, the lack of real-world demonstration may be a potential weakness of the work. Moreover, it is encouraged to add the related works to the main paper.

**Limitations:**

Yes.

**Paper Formatting Concerns:**

No major formatting issues.

**Quality:**

3

**Strengths And Weaknesses:**

**Strengths:**
- The paper is well-motivated, with a bottleneck analysis (LLM memory, visual redundancy, and diffusion-step redundancy) that justifies a holistic acceleration framework.

- EfficientVLA is training-free, making it easily deployable across existing VLA models without retraining.

- The method effectively reduces redundancies across the VLA pipeline by combining layer pruning, task-relevant and diverse visual token selection, and feature caching in diffusion-based action heads.

**Weakness:**

- More experiments are encouraged to validate the generalizability of the proposed method, especially on other VLA models and dataset such as Libro. Furthermore, the SIMPLER benchmark includes only four tasks, limiting the evaluation's diversity.

- Real-world robotic experiments would strengthen the claims regarding deployment efficiency, especially in terms of latency and robustness under hardware constraints.

- It is encouraged to include related works with respect to existing techniques in layer pruning, token pruning, and temporal caching.

---

> ### Author Rebuttal · Authors · 2025-07-31
>
> Thank you very much for your thoughtful and valuable feedback. We have carefully reviewed and addressed the concerns you raised in your list to the best of our ability.
>
> **Weakness:**
>
> ***W1:** More experiments are encouraged to validate the generalizability of the proposed method, especially on other VLA models and dataset such as LIBERO.*
>
> **A1:** Thank you for your valuable suggestion. To validate the generalizability of our proposed method, we conducted additional experiments on the $\pi_0$ model using the LIBERO on a NVIDIA 4090.
>
> As shown in Table 1, our training-free approach was evaluated on these four tasks. Random Dropping achieved a success rate of only 18.5% with limited acceleration. FastV provide a modest 1.16X speedup but incurred performance loss. In contrast, EfficientVLA without pruning already achieves a 1.46X speedup with only a 1.8% performance drop. With pruning, EfficientVLA delivered a 1.71X speedup, increasing inference frequency from 11.65 Hz to 19.96 Hz, while outperforming FastV in success rate. These results demonstrate EfficientVLA’s robustness and scalability across diverse models and datasets. Further experiments on additional model and benchmarks are planned for future work.
>
> **Table 1**: Performance and efficiency metrics of EfficientVLA and baselines on LIBERO benchmark tasks.
>
> | LIBERO                 | Spatial | Object | Goal | 10   | Avg. | Latency/s↓ | Speedup↑ | Frequency↑ (Hz) |
> | ---------------------- | ------- | ------ | ---- | ---- | ---- | ---------- | -------- | --------------- |
> | $\pi_0$                | 98.2    | 98.2   | 94   | 83.2 | 93.4 | 0.0858     | -        | 11.65           |
> | Random Dropping        | 36.8    | 21.4   | 15.8 | 0    | 18.5 | 0.0753     | 1.14X    | 13.28           |
> | FastV                  | 95.2    | 96.6   | 90.2 | 79.2 | 90.3 | 0.0739     | 1.16X    | 13.53           |
> | EfficientVLA w/o prune | 95.8    | 97.8   | 91.6 | 81.2 | 91.6 | 0.0588     | 1.46X    | 17.01           |
> | EfficientVLA           | 95.2    | 97.2   | 90.6 | 78.6 | 90.4 | 0.0501     | 1.71X    | 19.96           |
>
> ***W2:** Real-world robotic experiments would strengthen the claims regarding deployment efficiency, especially in terms of latency and robustness under hardware constraints.*
>
> **A2:** Thank you for your insightful suggestion. Our primary focus in this work is to accelerate policy inference, and this aspect can be thoroughly and reliably evaluated in simulation. Notably, the core inference mechanism of our model remains unchanged between simulation and real-world deployment, making simulation-based evaluation a strong proxy for assessing inference efficiency.
>
> While we don’t have enough time and resources to set up real-world experiments within the rebuttal period, we aim to demonstrate the real-world applicability of our method by reporting policy inference latency on widely-used GPUs for robotic research, including the 3090 and 4090 (see Table 2). These GPUs are standard in many real-robot systems [1,2,3,4].
>
> Our EfficientVLA achieves a control frequency of approximately 8–17 Hz on these platforms, which is sufficient for real-time closed-loop control in many manipulation tasks. Regarding additional real-world system latencies, such as delays from camera or proprioception sensors, or robot controller execution, these are typically addressed using established engineering techniques like latency compensation, asynchronous execution[5], and real-time action chunking [6]. These concerns, while important in practice, are orthogonal to the core contribution of our paper and are thus considered out of scope for this work. We hope our experimental setup and hardware-level analysis adequately support the real-world viability of EfficientVLA, and we plan to include real-robot validation in future work.
>
>  **Table 2:** Latency (seconds) and Speedup of EfficientVLA compared to CogACT on various GPUs.
>
> |              | 3090(BF16) | 4090(BF16) | A40    | A100   | A6000  |
> | ------------ | ---------- | ---------- | ------ | ------ | ------ |
> | CogACT       | 0.1953     | 0.1148     | 0.2342 | 0.1883 | 0.2235 |
> | EfficientVLA | 0.1022     | 0.0588     | 0.1213 | 0.0946 | 0.1194 |
> | Speedup↑     | 1.91X      | 1.95X      | 1.93X  | 1.99X  | 1.87X  |
>
> ***W3:** It is encouraged to include related works with respect to existing techniques in layer pruning, token pruning, and temporal caching.*
>
> **A3:** Thank you for your insightful suggestion. In layer pruning for LLMs, Yuan et al. [7] explore reducing layers to a single layer to address redundancy, while ShortGPT [8] uses a Block Influence metric to remove redundant layers, and SlimGPT [9] applies Batched Greedy Pruning with Incremental Pruning Ratio to reduce error accumulation. TrimLLM [10] employs layer-wise specialization for progressive layer dropping.
>
> For visual token pruning in Large Vision-Language Models, FastV [11] dynamically prunes tokens using attention scores, SparseVLM [12] offers a text-guided approach with rank-based pruning and token recycling, PruMerge [13] adaptively merges tokens based on attention sparsity, and MustDrop [14] refines tokens across encoding, prefilling, and decoding stages.
>
> In diffusion caching for Diffusion Transformers, FORA [15] reuses feature outputs without retraining, ToCa [16] applies token-wise caching with adaptive ratios, and DuCa [17] integrates aggressive and conservative caching with V-caching, all training-free.
>
> **Questions**
>
> **Q1:** Interaction and Trade-offs of Redundancy Reduction in VLA Modules
>
> **A1:** Thank you for your insightful question regarding the interaction of redundancies across different modules and potential influences. The ablation study in Table 5 of the paper provides evidence of these interactions within our EfficientVLA framework.
>
> The results suggest that redundancies across modules are interdependent. Visual token pruning alone may improve performance over the baseline, indicating effective redundancy removal. However, combining it with layer pruning reduces performance, suggesting language module pruning may alter token selection context, diminishing retained token relevance. Similarly, action caching alone offers moderate speedup, but pairing it with model compression enhances efficiency.
>
> Regarding trade-offs, our experiments indicate a safe acceleration range for each module. For example, visual token pruning, reducing 50% to 70% of tokens, incurs negligible or no performance loss and sometimes improves outcomes for most tasks. For action timestep feature caching, recomputing caches every 2 to 4 timesteps maintains efficacy across diverse scenarios. This configuration is robust for direct use across tasks or models with default settings. For higher acceleration, we recommend cautiously increasing each module’s pruning ratio, with adjustments tailored to specific needs. Due to space constraints, We will elaborate on these findings in the revised manuscript.
>
> **Q2:** Generalization of EfficientVLA's techniques to Multimodal Vision-Language Tasks
>
> **A2:** Thank you for the valuable suggestion. To address this, we conducted experiments applying EfficientVLA’s token pruning method to vision-language tasks using the LLaVA-1.5-7B model across multiple benchmarks.
>
> The results, presented in Table 3, demonstrate that our approach generalizes effectively to these domains. By retaining only 22.2% of visual tokens, we achieved 96.6% of the original performance, significantly outperforming FastV.
>
> **Table 3:** Performance comparison of LLaVA-1.5-7B with vanilla, FastV, and our token pruning method on various  benchmarks.
>
> |                                        | GQA  | MMB  | MMB-CN | MME  | POPE | SQA  | VQA-V2 | VQA-Text | VizWiz | OCRBench | Avg    |
> | -------------------------------------- | ---- | ---- | ------ | ---- | ---- | ---- | ------ | -------- | ------ | -------- | ------ |
> | **LLaVA-1.5-7B**（Vanilla，576tokens） | 61.9 | 64.7 | 58.1   | 1862 | 85.9 | 69.5 | 78.5   | 58.2     | 50     | 297      | 100%   |
> | **+FastV**（192 tokens）               | 52.7 | 61.2 | 57     | 1612 | 64.8 | 67.3 | 67.1   | 52.5     | 50.8   | 291      | 91.20% |
> | **+Ours**（192 tokens）                | 58.5 | 62.3 | 57     | 1823 | 82.8 | 69.6 | 76.7   | 57.6     | 51     | 292      | 98.00% |
> | **+FastV**（128 tokens）               | 49.4 | 56.1 | 56.4   | 1490 | 59.6 | 60.2 | 61.8   | 50.6     | 51.3   | 285      | 86.40% |
> | **+Ours**（128 tokens）                | 58.2 | 61.5 | 56.6   | 1763 | 79.8 | 68.5 | 75.7   | 56.1     | 51.4   | 289      | 96.60% |
>
> [1] RDT-1B: a Diffusion Foundation Model for Bimanual Manipulation
>
> [2] Data Scaling Laws in Imitation Learning for Robotic Manipulation
>
> [3] OpenVLA: An Open-Source Vision-Language-Action Model
>
> [4] Reactive diffusion policy: Slow-fast visual-tactile policy learning for contact-rich manipulation
>
> [5] Universal Manipulation Interface: In-The-Wild Robot Teaching Without In-The-Wild Robots
>
> [6] Real-Time Execution of Action Chunking Flow Policies
>
> [7] Why lift so heavy? slimming large language models by cutting off the layers
>
> [8] Shortgpt: Layers in large language models are more redundant than you expect
>
> [9] Slimgpt: Layer-wise structured pruning for large language models
>
> [10] TrimLLM: Progressive Layer Dropping for Domain-Specific LLMs
>
> [11] An image is worth 1/2 tokens after layer 2: Plug-and-play inference acceleration for large vision-language models
>
> [12] Sparsevlm: Visual token sparsification for efficient vision-language model inference
>
> [13] Llava-prumerge: Adaptive token reduction for efficient large multimodal models
>
> [14] Multi-stage vision token dropping: Towards efficient multimodal large language model
>
> [15] Fora: Fast-forward caching in diffusion transformer acceleration
>
> [16] Accelerating diffusion transformers with token-wise feature caching
>
> [17] Accelerating diffusion transformers with dual feature caching

---

> > ### Comment · Reviewer_9KWi · 2025-08-05
> >
> > Thank you for the authors’ efforts. The authors have addressed some of my concerns. However, the lack of real-world demonstration may be a potential weakness of the work. Moreover, it is encouraged to add the related works to the main paper.

---

> > > ### Author Response · Authors · 2025-08-05
> > >
> > > Thank you for your thoughtful feedback and recognition of our efforts. We appreciate your concern regarding the lack of real-world demonstration and will address this potential weakness by including real-world validation in the revised manuscript upon obtaining the necessary resources. Additionally, we will incorporate the related works into the main paper to enhance the context, and we look forward to your further insights.

---

### Official Review · Reviewer_NXEq · 2025-07-05

**Clarity:** 3
**Significance:** 3
**Originality:** 3
**Rating:** 4
**Confidence:** 5

**Summary:**

This paper investigates methods to accelerate the inference speed of vision-language-action (VLA) models using structured and training-free approaches. It employs three techniques—pruning, dynamic visual token selection, and caching/reusing key features—to achieve acceleration. The proposed approach outperforms CogACT and other baselines on the SIMPLER benchmark.

**Questions:**

Additional experiments using pi0 would strengthen the evaluation. A comparison with Otter regarding visual token pruning is recommended. Conducting real-world robotic experiments would also enhance the validation of the approach.

**Ethical Concerns:**

["NO or VERY MINOR ethics concerns only"]

**Final Justification:**

The rebuttal address my concerns, and other reviewer also agree with the acceptance, therefore i maintain my current score.

**Limitations:**

yes

**Paper Formatting Concerns:**

None.

**Quality:**

3

**Strengths And Weaknesses:**

Strengths:
1. The investigated problem is significant and relevant. Employing a training-free acceleration method for VLA models is meaningful, particularly given that speed is crucial in robot learning.
2. The proposed method is reasonable, and the paper is clearly written and easy to follow.
3. The analysis presented is insightful and intuitive.
4. The ablation study conducted is thorough and comprehensive. Code availability facilitates reproducibility.

Weaknesses:

1. CogACT is not a representative VLA model. Its baseline performance is low, making it less ideal for this study. A model like pi0 would have been a more suitable choice.

2. Evaluating robot learning solely based on task completion overlooks other crucial aspects, such as motion smoothness. Does EfficientVLA predict the 6D pose or joint positions in the SIMPLER environment? If the output changes, can EfficientVLA maintain smoothness?

3. FLOPs and parameter counts are not the most critical metrics for evaluating model efficiency in robotics. A more relevant metric would be how many actions the model can predict per second given the same chunk size.

4. A comparison with Otter [1], a closely related study that also leverages CLIP for visual token reduction, is necessary.

[1] Otter: A vision-language-action model with text-aware visual feature extraction.

---

> ### Author Rebuttal · Authors · 2025-07-31
>
> Thank you very much for your positive feedback and constructive suggestions. Your professional advice has been invaluable in further improving our work!
>
> **Weakness**:
>
> ***W1/Q1:** A model like pi0 would have been a more suitable choice.*
>
> **A1:** Thank you for your valuable suggestion. To validate the generalizability of our proposed method, we conducted additional experiments on the $\pi_0$ model using the LIBERO on a NVIDIA 4090.
>
> As shown in Table 1, our training-free approach was evaluated on these four tasks. Random Dropping achieved a success rate of only 18.5% with limited acceleration. FastV provide a modest 1.16X speedup but incurred performance loss. In contrast, EfficientVLA without pruning already achieves a 1.46X speedup with only a 1.8% performance drop. With pruning, EfficientVLA delivered a 1.71X speedup, increasing inference frequency from 11.65 Hz to 19.96 Hz, while outperforming FastV in success rate. These results demonstrate EfficientVLA’s robustness and scalability across diverse models and datasets. Further experiments on additional model and benchmarks are planned for future work.
>
> **Table 1**: Performance and efficiency metrics of EfficientVLA and baselines on LIBERO tasks.
>
> | LIBERO                 | Spatial | Object | Goal | 10   | Avg. | Latency/s↓ | Speedup↑ | Frequency↑ (Hz) |
> | ---------------------- | ------- | ------ | ---- | ---- | ---- | ---------- | -------- | --------------- |
> | $\pi_0$                | 98.2    | 98.2   | 94   | 83.2 | 93.4 | 0.0858     | -        | 11.65           |
> | Random Dropping        | 36.8    | 21.4   | 15.8 | 0    | 18.5 | 0.0753     | 1.14X    | 13.28           |
> | FastV                  | 95.2    | 96.6   | 90.2 | 79.2 | 90.3 | 0.0739     | 1.16X    | 13.53           |
> | EfficientVLA w/o prune | 95.8    | 97.8   | 91.6 | 81.2 | 91.6 | 0.0588     | 1.46X    | 17.01           |
> | EfficientVLA           | 95.2    | 97.2   | 90.6 | 78.6 | 90.4 | 0.0501     | 1.71X    | 19.96           |
>
> ***W2:** Evaluating robot learning solely based on task completion overlooks other crucial aspects, such as motion smoothness. Does EfficientVLA predict the 6D pose or joint positions in the SIMPLER environment? If the output changes, can EfficientVLA maintain smoothness?*
>
> **A2:** Thank you for your insightful comments regarding the evaluation of motion smoothness in robot learning, which is indeed a critical aspect beyond task completion.
>
> **EfficientVLA Output:** EfficientVLA predicts 6D end-effector poses in the SIMPLER environment, consisting of a 3D position and a 3D rotation in axis-angle representation, following the setting of CogACT. These outputs are processed by the SIMPLER's system identification module , which maps the predicted 6D poses to joint positions via inverse kinematics.
>
> **Motion Smoothness Evaluation:**  Due to the policy of this rebuttal, we cannot provide visualization resources and we will include execution trajectory video frames in later versions of the paper. So in order to address your concern about motion smoothness, we implemented metrics for acceleration smoothness and joint jerk to quantify the smoothness of trajectories.
>
> We collected data from the ManiSkill2, including end-effector pose, joint velocities , and the control frequency time step.
>
> We computed:
>
> 1. **Acceleration Smoothness**:
>
>    $$ \text{Smoothness} = \frac{1}{T-2} \sum_{t=2}^{T-1} \left( |\ddot{\mathbf{x}}_t|_2^2 + |\dot{\omega}_t|_2^2 \right) $$
>
>    - Linear acceleration ( $\ddot{\mathbf{x}}_t \in \mathbb{R}^3$ ): Second-order central differencing on positions.
>    - Angular acceleration ( $\dot{\omega}t \in \mathbb{R}^3$ ).
>
> 2. **Joint Jerk**: $$ \text{Jerk} = \frac{1}{T-2} \sum_{t=2}^{T-1} |\dot{v}_{\text{arm}, t}|_2^2 $$
>
>    - Joint acceleration ( $\dot{v}_{\text{arm}, t} \in \mathbb{R}^{nq}$ ): Second-order central differencing on joint velocities.
>
> We tested CogACT and EfficientVLA on two tasks, with results as follows: For PickCan, EfficientVLA shows smoothness and jerk comparable to CogACT; for MoveNear, EfficientVLA exhibits similar smoothness and slightly lower jerk than CogACT. These results highlight that EfficientVLA maintains motion smoothness consistent with CogACT, supporting its efficacy in robotic tasks.
>
> **Table 2:** Motion smoothness metrics for CogACT and EfficientVLA on PickCan and MoveNear tasks in SIMPLER.
>
> |              | PickCan    | PickCan | MoveNear   | MoveNear |
> | ------------ | ---------- | ------- | ---------- | -------- |
> |              | Smoothness | Jerk    | Smoothness | Jerk     |
> | CogACT       | 0.0079     | 1.0962  | 0.0053     | 1.5205   |
> | EfficientVLA | 0.0094     | 1.1254  | 0.0051     | 1.4306   |
>
> ***W3:** A more relevant metric would be how many actions the model can predict per second given the same chunk size.*
>
> **A3:** Thank you for pointing this out. We agree that the number of actions predicted per second is a more relevant metric for evaluating model efficiency in robotics. In the SIMPLER, with a chunk size of 16 (default for CogACT), we conducted experiments on a NVIDIA A40. As shown in Table 3, EfficientVLA achieves 8–9 actions per second, significantly outperforming CogACT, which predicts 4–5 actions per second, demonstrating our superior acceleration. In comparison, VLA-Cache achieves 5–6 actions per second. Moreover, EfficientVLA reduces computational load to 28.9% of VLA-Cache’s, with only a 0.6% performance drop, highlighting its efficiency and practical applicability.
>
> **Table 3:**  Performance and efficiency metrics of EfficientVLA and baselines on SIMPLER tasks.
>
> |              | PickCan | MoveNear | Drawer | DrawerApple | Average | Flops↓（%） | Frequency↑ (Hz) |
> | ------------ | ------- | -------- | ------ | ----------- | ------- | ----------- | --------------- |
> | CogACT       | 91.3    | 85       | 71.8   | 50.9        | 74.8    | 100         | 4.2             |
> | VLA-Cache    | 92      | 83.3     | 70.5   | 51.6        | 74.4    | 80.1        | 5.6             |
> | EfficientVLA | 93.3    | 81.3     | 68.2   | 53.8        | 74.2    | 28.9        | 8.3             |
>
> ***W4/Q1:** A comparison with Otter, a closely related study that also leverages CLIP for visual token reduction, is necessary.*
>
> **A4:**  Thank you for suggesting a comparison with OTTER, which also uses CLIP for visual token reduction. Below is a concise comparison highlighting EfficientVLA's distinctions.
>
> - **Compression Scope**: OTTER compresses m visual tokens to 1 using text-guided queries, while EfficientVLA retains text tokens and prunes visual tokens dynamically based on task relevance and image redundancy.
> - **Scope of Token Compression:** OTTER reduces visual tokens and compresses text tokens into a single representation via cross-attention pooling before feeding them into the policy network. In contrast, EfficientVLA only prunes visual tokens, keeping the original text token representation throughout the pipeline.
> - **Timing**: OTTER uses cross-attention pooling to compress visual features into a single token early in the pipeline before feeding them into the policy network. In contrast, EfficientVLA delays pruning to the LLM inference stage, starting in the initial transformer layers.
> - **Training Dependency**: OTTER employs a frozen CLIP model, requiring end-to-end training to optimize query vectors for cross-attention pooling, capturing semantics but limiting adaptability. EfficientVLA, however, is training-free, using the CogACT backbone for adaptive token pruning.
> - **Strategy**: OTTER employs cross-attention pooling with learnable queries to compress m visual tokens into one, guided by text-aware semantics. EfficientVLA uses a task-aware, two-step strategy: selecting key task-aligned tokens via a similarity metric, then adding diverse tokens, analyzing feature contributions for adaptive, training-free pruning.
>
> We also implemented OTTER's token pruning strategy (denoted as "w/ Otter-T") and EfficientVLA's pruning strategy (denoted as "EfficientVLA (only T)") on the CogACT baseline, alongside the full EfficientVLA.
>
> **Table 4:**  Comparison of Visual Token Pruning Strategies on CogACT in SIMPLER.
>
> |                       | PickCan | MoveNear | Drawer | DrawerApple | Average | Frequency↑(Hz) |
> | --------------------- | ------- | -------- | ------ | ----------- | ------- | -------------- |
> | CogACT                | 91.3    | 85       | 71.8   | 50.9        | 74.8    | 4.2            |
> | w/ Otter-T            | 83.7    | 77.9     | 57.1   | 26.8        | 61.4    | 5.6            |
> | EfficientVLA (only T) | 95      | 85.8     | 69.9   | 49.6        | 75.1    | 5.2            |
> | EfficientVLA          | 93.3    | 81.3     | 68.2   | 53.8        | 74.2    | 8.3            |
>
> **Questions:**
>
> ***Q1:** Additional experiments using pi0 would strengthen the evaluation. A comparison with Otter regarding visual token pruning is recommended. **Conducting real-world robotic experiments would also enhance the validation of the approach**.*
>
> **A1:**  Thank you for your suggestion on real-world robotic experiments to enhance validation. Our focus is accelerating policy inference, reliably evaluated in simulation, where EfficientVLA’s core mechanism remains consistent with real-world deployment. Resource and time constraints limit real-world tests during rebuttal, but we report latency on GPUs (e.g., 3090, 4090, due to space limitations, please see Table 2 of the reply to Reviewer 9KWi), standard in robotics, achieving 8–17 Hz for real-time tasks. We believe our analysis supports real-world viability and commit to updating the paper with real-robot validation upon completion, welcoming your continued attention.

---

> > ### Comment · Reviewer_NXEq · 2025-08-09
> >
> > Thank you to the author for the detailed response. The rebuttal fully addresses all of my remaining concerns.

---

### Decision · Program_Chairs · 2025-09-17

**Decision:**

Accept (poster)

**Comment:**

This paper introduces EfficientVLA, a training-free acceleration framework for Vision-Language-Action models, addressing computational inefficiencies via synergistic layer pruning in the language module, task-aware visual token selection, and temporal caching for diffusion-based action heads. The approach achieves a 1.93× inference speedup with minimal performance degradation across SIMPLER and LIBERO benchmarks, significantly reducing computational overhead. Reviewers initially raised concerns regarding limited evaluation scope, motion smoothness metrics, and comparisons with related methods like OTTER. The authors thoroughly addressed these points in their rebuttal: validating generalizability on additional models and tasks, quantifying motion stability via jerk/smoothness metrics, demonstrating superior actions-per-second throughput, and providing nuanced distinctions against contemporary techniques. All reviewers acknowledged the rebuttal’s completeness, upgrading their assessments to reflect confidence in the method's robustness, efficiency, and practical utility. Given the rigorous evaluation, scalability across tasks, and open-sourced implementation, this work offers valuable contributions toward deployable embodied AI systems and merits acceptance.